# Hydrochemical and Microbiological Investigations and the Therapeutic Potential of Some Mineral Waters from Bihor County, Romania

Ribana Linc [1,*,†] , Emilia Pantea [2,†] , Eugenia Șerban [2,†] , Anca-Paula Ciurba (Pastor) [1,3,†] and Georgeta Serban [4,†]

1   Department of Geography, Tourism and Territorial Planning, Faculty of Geography, Tourism and Sports, University of Oradea, 1 University St., 410087 Oradea, Romania; anca-paula.ciurba@univ-lorraine.fr
2   Faculty of Environmental Protection, University of Oradea, 26 Gen. Magheru St., 410057 Oradea, Romania; emilia.pantea@uoradea.ro (E.P.); eugeniaserban@yahoo.com (E.Ș.)
3   Laboratory LOTERR-EA 7304, Department of Geography, University of Lorraine, 57045 Metz, France
4   Faculty of Medicine and Pharmacy, University of Oradea, 29 Nicolae Jiga, 410028 Oradea, Romania; getaserban_2000@yahoo.com
*   Correspondence: ribanalinc@yahoo.com
†   All authors contributed equally to this work.

**Abstract:** Water quality plays an important role for every sustainable social and economic system, as well as for maintaining human health. This study, carried out during 2022–2023, has as its main objective the physical–chemical and microbiological analysis of some underground water resources (two boreholes and a spring) with a natural mineral load from three areas in Bihor County (Romania), and the impact of their consumption on people's health. Therefore, six microbiological parameters and 17 physical–chemical indicators in three localities (Tămășeu, Sîntimreu, Pădurea Neagră) were analyzed. The results of the microbiological analysis indicate a type of water that respects the limits imposed by the legislation on natural mineral waters and potability. The physical–chemical indicators show that the hydrochemical type of the studied waters is predominantly bicarbonate, in association with calcium, magnesium, and sodium cations. The residents' perception on water quality and the effect on people's health was assessed through a questionnaire (23 items) addressed to the population of the three villages and neighboring localities. The results showed that the mineral waters from Sîntimreu and Pădurea Neagră are used frequently. Many respondents consider the local mineral waters as without quality-related problems and with beneficial effects regarding acute or chronic gastrointestinal conditions, such as gastritis, gastric ulcers, flatulence, or liver diseases.

**Keywords:** natural mineral water; hydrochemistry; microbiological analysis; spring; borehole; water quality; therapeutic effect; diseases

## 1. Introduction

Water is one of the main sources of our existence on Earth. Alongside population growth and rising living standards, water consumption has also increased, which has led to a decrease in the resources available to ensure the daily requirement of fresh water. As a result, the identification of new water resources that can be the source of drinking water but can also bring benefits to human health is an important objective of the policies applied to this sector. Natural mineral waters might be valuable resources for the population of any country. Some authors [1,2] found a sudden increase in mineral water consumption worldwide, and that is supported by the very beneficial properties of these waters for human health [3]. According to [4], Romania has a diversified and significant range of natural mineral water resources for food consumption, and there is potential for their exploitation. The hydromineral sources are concentrated on the territory of 16 counties

(Figure 1): Arad, Arges, Bihor, Brașov, Caraș-Severin, Covasna, Harghita, Hunedoara, Iași, Maramureș, Mureș, Neamț, Prahova, Satu-Mare, Suceava, and Timiș [5].

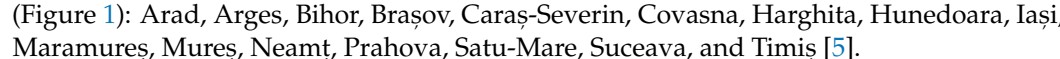

**Figure 1.** Bihor County: localities that have natural mineral water and carbon dioxide resources. In the inset—Romania and the counties that have natural mineral water resources and the number of sources recognized by the National Agency for Mineral Resources from Romania.

According to Feru [6], over 45% of natural mineral water deposits in Romania are located in carbonate rocks (limestone, calcareous conglomerates, etc.), about 25% are located in igneous rocks (pyroclastites and andesites), 25% in detritus sedimentary deposits, and 5% in sandstones and crystalline shales. Before the 1990s, the condition for mineral water to be bottled in Romania was that it had to contain a quantity of dissolved salts higher than

1000 mg/L, a carbon dioxide ($CO_2$) content higher than 500 mg/L, and have the therapeutic effects mentioned on the label. Nowadays, from Government Decision HG 1020/2005 [7] implementing Directive 80/777/EEC [8] (with subsequent additions), a distinction is made between the terms "natural mineral waters" and "medicinal waters", meaning that the medicinal waters must have scientifically recognized therapeutic actions, and can exceed the maximum allowed concentrations even for some undesirable and potentially toxic elements which have been proven to be responsible for the therapeutic effects (e.g., iron, manganese, arsenic, hydrogen sulphide). Teodoreanu and Gaceu [9] assume that 37% of Europe's mineral waters are owned by Romania, being spread over the three major relief steps (plains, hills, mountains) as springs and mineral borehole waters with different flow, temperatures, and chemical content.

Water has been used to promote health since ancient times. It was only in the 19th century that detailed chemical analyses brought information about mineral content, and thus revealed the medicinal potency and healing benefits of mineral waters in various pathologies [10,11]. Mineral waters have a beneficial effect on human health both through internal administration (crenotherapy) in the form of water ingestion or inhalation of aerosols, and through external administration (balneotherapy) by immersing the body in water [10,11]. Depending on their chemical content, mineral waters have been used for decades as curative agents in the treatment of various gastrointestinal conditions such as heartburn, dyspepsia, gastritis, ulcers, constipation, irritable bowel syndrome, and even gallbladder or liver diseases, as an adjuvant therapy to complement the effects of the drugs [11,12]. The reduced volume of urine or an increased urinary concentration of salts favors the formation of kidney stones. Crenotherapy is effective in the prevention and therapy of urolithiasis because a proper consumption of water washes the urinary tract, increases the urinary volume, increases the dilution of dissolved salts, and reduces microbial content and changes in urinary pH [13–15].

Crenotherapy has beneficial results in the therapy of respiratory pathologies. Steam or aerosol inhalation, as well as irrigation with high mineral content waters due to the several effects of minerals (antiseptic effect—sodium chloride, metabolic effects—iodine and iodides, analgesic and sedative effects—bromide and calcium) is an accessible method with long-term beneficial results for respiratory conditions such as allergies, general inflammation of the airways (chronic obstructive pulmonary disease—COPD), recurrent viral and bacterial infections, abnormalities in bronchial secretions, and impairment of lung and nasal functions [16]. Although the mode of action is still unclear, the anti-inflammatory, immunomodulatory, and mucolytic activities of mineral waters might make a substantial contribution to therapeutic effects, with the advantage of the lack of possible side effects of drugs usually used in the treatment of respiratory diseases (e.g., corticosteroids, antibiotics, bronchodilators). Inhalations with mineral water relieve the symptoms of lower and upper respiratory tract diseases, such as rhinitis, laryngitis, pharyngitis, rhinosinusitis, asthma, bronchitis, and pneumonia, and also improve patients' quality of life [12,16–18].

Sulphurous mineral waters in particular have multiple therapeutic effects. In addition to gastrointestinal and respiratory effects, since hydrogen sulfide is a small-sized gas molecule, it can be easily absorbed through the skin and mucosa, and gives systemic effects [19]. Thus, sulphurous waters are effective in conditions such as hypertension, heart disease, ischemia, or lower and upper urinary tract disorders [19–21]. During hydropinotherapy with sulphurous mineral waters, there is an increase in the secretion of insulin, leading to the burning of glucose [22]. In addition, a decrease in the concentration of reactive oxygen species was observed. The antioxidant properties are accompanied by anti-aging effects, as these mineral waters are able to prevent structural changes in DNA [22,23].

Bihor County is known in Romania as having significant resources of geothermal mineral waters that have been intensively studied [24–28], but there are few studies on the analysis of *cold natural mineral waters* [29–31]. Considering this fact and in correlation with the current trends regarding the sustainable management of water resources, the

main purpose of this work is to provide the physical–chemical (pH, dry residue, chemical composition) and microbiological characteristics of some cold natural mineral waters from three sources in Bihor County (Tămășeu, Sîntimreu, Pădurea Neagră). Taking into account the consumption of these waters by the population of the respective areas, the first and second hypotheses of this study are as follows:

**Hypothesis 1.** *The physical characteristics and chemical composition of the water is a condition for its inclusion in the category of mineral water.*

**Hypothesis 2.** *The microbiological purity of mineral water is a condition for being admitted to human consumption.*

Since old documents cited by [32] suggest that lithium was the reason for the use and commercialization of mineral water from the Tămășeu for about 60 years, the third hypothesis is deduced as follows:

**Hypothesis 3.** *The lithium content is a condition for the mineral water from the Tămășeu borehole being classified as lithiniferous water.*

In addition, in order to highlight the potential impact of the consumption of these mineral waters on people's health, we aimed to find out the population's perception towards the use of water from the three sources. Therefore, a questionnaire addressed to the resident population of the three locations, as well as the neighboring villages and towns (totaling 12 localities), was studied and applied. A fourth hypothesis is deduced as follows:

**Hypothesis 4.** *Population awareness of the quality and possible therapeutic effects of the mineral waters from the three sources.*

To our knowledge, there are very few and old studies analyzing the water from the three sources [33–36]; therefore, our study brings current knowledge about the physical–chemical and microbiological characteristics of these waters, as well as their possible further exploitation for nutritional and therapeutic purposes. This information is useful to both the scientific and local communities.

## 2. Materials and Methods

### 2.1. Study Area

Located in northwestern Romania, Bihor County has various underground resources of recognized economic importance, such as coal and crude oil deposits concentrated especially in the Barcău river basin, as well as important geothermal water deposits spread throughout the county. Also concentrated in the Barcău basin, there are *several springs of cold natural mineral waters and resources of carbon dioxide.*

Due to $CO_2$ coming from deep in the fault lines of the foundation, in the *Western Plain of Bihor County,* there are several locations with bicarbonate water springs north of the Oradea city (at Sălard, Sîntimreu, Tămășeu), as well as in the Tinca resort located in the south of the county (here being exploited for therapeutic purposes). In the *Bihorului Mountains*, there are still natural mineral waters in the Budureasa commune (Izvorul Minunilor from the Stâna de Vale resort and Izvorul Cuciului/Hera), found on the annual list of recognized mineral waters in Romania [37]. There is also a spring in Pădurea Neagră holiday village, located in the *Plopisului Mountains*, from where people frequently supplied themselves with water for individual consumption.

In this study, we focused on three easily accessible sources of *cold natural mineral water*: two medium-depth boreholes in the plain area at **Tămășeu** and **Sîntimreu,** and a spring in the mountaineous area of the **Pădurea Neagră** holiday village (Figure 1).

These sources have been known for a long time, and those in Tămășeu and Sîntimreu were used by the locals for their therapeutic properties especially in the late 19th–early 20th

century. The boreholes were less used after the Second World War, so the locals now know little about the properties and qualities of these waters. However, until Romania joined the EU in 2007, these waters were used for individual consumption, but the opening of many hypermarkets that sell various types of water at relatively low prices, as well as the fact that these waters acquire a reddish color after a while, have considerably reduced the supply flow. Currently, the locals or people passing through the area continue to supply themselves even sporadically with natural mineral water from the boreholes in Tămășeu and Sîntimreu and the spring in Pădurea Neagră. Only on very hot summer days, a large number of people can be seen stocking up on water for consumption.

The benchmark of the study is the specifications of Directive 54/2009 [38], according to which natural mineral water can be differentiated from ordinary drinking water: *(a)* by its nature, characterized by its mineral components, oligoelements or other constituents and, possibly, by certain therapeutic effects; *(b)* by its original purity, both characteristics being kept unchanged due to the underground origin of the mineral water, which has been protected against all risks of pollution. The basic national legislation that constituted the point of reference was Government Decision HG 1020/2005 [7], supplemented and amended by HG 532/2010 [39], the main normative act that legislates the exploitation and commercialization of natural mineral waters in Romania.

### 2.2. Determination of Physical-Chemical Characteristics

The characteristics of mineral waters are the result of the mineralogical composition of the rocks crossed by the water, the crossing time, the type of waters encountered, the mixing rate, as well as the duration of this process [40].

### 2.2.1. Sampling of Natural Mineral Waters

In order to achieve the objectives of this study, two sampling sessions were carried out from the three locations in April 2022 and May 2023, respectively. Polyethylene terephthalate (PET) bottles of two liters were used to determine the physical–chemical indicators. The samples were analyzed within 24 h from sampling. To determine the microbiological characteristics, sterilized glass flasks of 0.5 L with glass stoppers were used. Sampling for microbiological analyses was carried out in accordance with Romanian Standard SR EN ISO 19458/2007—Water quality [41]. Figure 2 shows the sampling points related to each water source analyzed in this study.

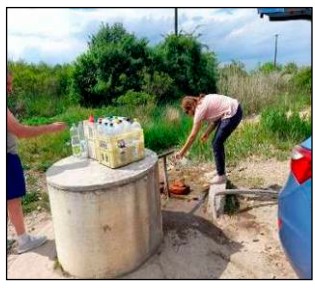 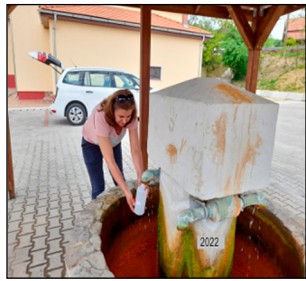 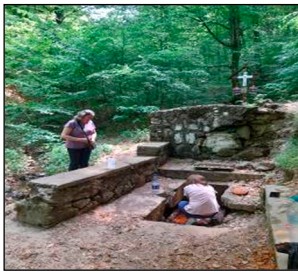

Tămășeu                                    Sîntimreu                                    Pădurea Neagră

**Figure 2.** The natural mineral water boreholes from Tămășeu and Sîntimreu, and the spring from Pădurea Neagră holiday village. Water sampling.

### 2.2.2. The Methodology for Determining the Main Physical–Chemical Indicators

A variety of working techniques—specific to each parameter analyzed for the three mineral water sources—were used in this study. The qualitative evaluation was carried out by comparing the data obtained with the limits established by the national/international legislation. The quality of chemical analyses was ensured by the use of high purity reagents (Merck, Bucharest, Romania), experimental methods approved by standardization bodies,

and laboratory equipment certified for high-precision chemical analyses. The main physical–chemical indicators analyzed are presented in Table 1.

**Table 1.** The main physical–chemical indicators.

| Indicators/U.M. | Methods of Analysis | Reference Documents |
|---|---|---|
| *Determination of pH* | Water quality. Determination of pH (Hanna Instruments HI2020-02 multiparameter, Hanna Instruments, Nusfalau, Romania) | [42] |
| *Conductivity (µS/cm)* | Water quality. Determination of electrical conductivity (Conductometer HI6321, Hanna Instruments, Nusfalău, Romania) | [43] |
| *Calcium, Magnesium (mg/L)* | Water quality. Determination of calcium and magnesium content by flame atomic absorption spectrometry (Perkin Elmer Analyst 800, Perkin Elmer LLC, Rodgau, Germany) | [44] |
| *Potassium, Sodium, (mg/L)* | Water quality. Determination of sodium and potassium. Part 3. Determination of sodium and potassium by flame emission spectrometry (Perkin Elmer Analyst 800, Perkin Elmer LLC, Rodgau, Germany) | [45] |
| *Lithium (mg/L)* | Water quality. Determination of selected elements by inductively coupled plasma optical emission spectrometry (ICPE-9800, Shimadzu, Shimadzu Europa GmbH, Duisburg, Germany) | [46] |
| *Iron (mg/L)* | Water quality. Determination of iron content by flame atomic absorption spectrometry (Perkin Elmer Analyst 800, Perkin Elmer LLC, Rodgau, Germany) | [47] |
| *Manganese (mg/L)* | Water quality. Determination of manganese content by flame atomic absorption spectrometry (Perkin Elmer Analyst 800, Perkin Elmer LLC, Rodgau, Germany) | [48] |
| *Chloride (mg/L)* | Water quality. Determination of chloride content. Volumetric (Mohr Method) | [49] |
| *Sulfate (mg/L)* | Water quality. Turbidimetric determination of sulphates | [50] |
| *Nitrates (mg/L)* | Water quality. Determination of nitrate. Part 3. Spectrometric method using sulfosalicylic acid (HACH DR 3900 spectrophotometer—Hach, Romania) | [51] |
| *Nitrites (mg/L)* | Water quality. Determination of nitrites. Molecular spectrometric absorption method (HACH DR 3900 spectrophotometer—Hach, Romania) | [52] |
| *$HCO_3^-$ (mg/L)* | Water quality. Determination of alkalinity. Part 2. Determination of carbonate alkalinity (volumetric method) | [53] |
| *$CO_2$ (mg/L)* | Water quality. Determination of carbon dioxide (Volumetric method) | [54] |
| *$H_2S$ (mg/L)* | Water quality. Determination of sulphide content (Iodometric method) | [55] |
| *Dry residue at 180 °C (mg/L)* | Water, groundwater and wastewater. Determination of residue. Gravimetric method (Biobase BOV-T30C oven, Biobase, China) | [56] |

Tools such as Piper and Stiff diagrams were used to process the analytical data obtained through the chemical analyses and to highlight the hydrochemical footprint of the samples [57].

*2.3. The Methodology for Determining the Microbiological Load*

Microbiological analysis of natural mineral water at the source can identify the presence of germs that can form colonies, on suitable culture media. This is particularly important for consumers, because after bottling, the number of germs can increase rapidly, reaching 104–105 CFU/mL (CFU = Colony Forming Unit) in 3–7 days, according to some authors [58]. Taking this into account, and correlated with the microbiological requirements provided by Government Decision HG 1020/2005 [7], supplemented and amended by HG 532/2010 [39], the microbiological study aimed to determine the microbiological parameters presented in Table 2.

**Table 2.** The main microbiological indicators.

| Type of Analysis | Methods of Analysis | Reference Documents/ Work Technique |
| --- | --- | --- |
| *Determination of the total colony count (CFU) at 22 °C* | Water quality. Enumeration of culture microorganisms. (Colony counting by seeding in agar-agar culture medium at a temperature of 22 °C, incubated for 72 h). | [59] |
| *Determination of the total colony count (CFU) at 37 °C* | Water quality. Enumeration of culture microorganisms. (Colony counting by seeding in agar-agar culture medium at a temperature at 37 °C incubated for 24 h). | [59] |
| *Determination of Coliform bacteria and isolation of Escherichia coli species* | Water quality. Enumeration of *Escherichia coli* and Coliform bacteria. Part 1: Membrane filtration method. | [60] |
| *Determination of intestinal Enterococci* | Water quality. Detection and enumeration of intestinal enterococci. Part 2: Membrane filtration method. (The determination was made after incubating the membrane for 48 h at 36 °C, on Slanetz–Bartley medium). | [44] |
| *Pseudomonas aeruginosa* | Water quality. Detection and enumeration of *Pseudomonas aeruginosa*. Membrane filtration method. (Pseudomonas agar/CN—agar medium membrane incubated at 36 °C for 49 h). | [61] |
| *Clostridium perfringens* | Water quality. Enumeration of *Clostridium perfringens*. Membrane filtration method. | [62] |

### 2.4. Questionnaire Content

The objective of the questionnaire was to assess the residents' perception on the quality and effect on health of the mineral waters from the three studied water sources (Tămășeu, Sîntimreu, and Pădurea Neagră). Therefore, a questionnaire addressed to the population residing in the three villages and neighboring localities (12 localities in total) was applied.

The questionnaire was divided into two parts. The first part included 18 items related to mineral waters, through which it was sought to obtain two types of data: one on the quality of mineral waters and the other on their possible effect on human health. The second part of the questionnaire included five items related to the socio-demographic situation of the respondents (residence, age, education, occupation etc.).

The 18 items had as main elements: identification of the main source of drinking water of the residents; identification of problems affecting the quality of mineral waters; consumption of mineral water by residents (amount, frequency, duration); identification of acute or chronic conditions of the population improved by mineral water consumption; identification of the possible beneficial effects on health reported after drinking mineral water; the use of mineral water for other purposes (e.g., in the kitchen).

The questionnaire included multiple-choice questions, as well as open-ended questions. It contained a standard, concise introduction stating the confidentiality of answers and their exclusive purpose for scientific research. Each respondent was asked for his or her willingness to participate in the study. All persons over 15 years old of age who agreed to answer the questionnaire were included in this study.

The questionnaire was prepared and pre-tested in a selected community. Thus, 15 questionnaires were distributed to the respondents face-to-face, to find out if the items were clearly expressed. After reviewing some items, the questionnaire was distributed to the population between June and July 2023. It was applied online, using the Google Forms application (online forms and surveys creator), where a link was created that was distributed to the respondents via Internet. Since many inhabitants of the villages of Tămășeu, Parhida, Sîntimreu and Sălard are of Hungarian ethnicity, the questionnaire was distributed both in Romanian and Hungarian for better understanding by the respondents. The questionnaire was also applied face-to-face, although the number of respondents was smaller. The face-to-face investigation was done by the authors of this study and made them better understand the problems and the real perception of the population on the quality and effect of mineral waters on human health.

110 questionnaires were returned from respondents, of which 102 were validated (92.7%). Respondents of different ages, sexes, and education levels, who were randomly chosen in the field, took part in the survey. The data collected from the 23 items were statistically processed with the Google Forms application that generates automatic summaries based on percentage calculation.

Expected impact: through the present survey, we aimed to raise the awareness of the population about the physical–chemical characteristics and therapeutic potential of the mineral waters in the studied area.

The method of using the questionnaire to highlight the population's perception on a phenomenon is often used in specialized literature [63–67].

## 3. Results and Discussion

### 3.1. Research Background

The spring from Pădurea Neagră, in the Plopișului Mountains, is found in the outskirts of Popești commune, in the east of the holiday village, in a mixed hornbeam and beech forest, and flows towards the Bistra river. Mihãilã [68] explains the formation of the mineral spring. Aniței et al. [36] mentioned this spring as having oligomineral waters and Nicula [69] notes the high concentration of iron due to the host rocks and the presence of $CO_2$, which increases the water's ability to dissolve minerals.

Throughout time, the cold natural mineral waters of the bihorean plain have been used for both consumption (Tămășeu, Sîntimreu, Tinca) and for balneary treatment and crenotherapy (Tinca, Tămășeu, Sălard). Following the disappearance of some springs due to the flooding of the Barcău River (Sălard), the frequent change of owners (Tămășeu) [32], the decrease in flow rates (Tămășeu), and the tightening of legislative conditions for bottling and marketing natural mineral waters, bottling was abandoned.

In the *Tămășeu—Sîntimreu—Sălard sector*, the hydromineral deposit contains bicarbonated, magnesian waters with a $CO_2$ content of over 1.5 g/L. The mineralized aquifer layer from the polygenic sands was opened at a depth of 120 m, and flows freely with low flow rates (below 0.2 L/s). In the 1970s, this aquifer was exploited by the bottling station in the Tămășeu railway station [33].

According to Szabó [32], who cites many Hungarian documents about the natural mineral water from Tămășeu, the Hungarian press wrote in 1899 about the drilling of the well in Tămășeu and the eruption of a "highly carbonated sour water", as well as about the chemical analyses carried out at the Institute of Chemistry in the Budapest II University of Sciences. The way to promote Tămășeu mineral water—called "*Lithium healing water from Tămășeu*"—through an intense popularization campaign, was also detailed. In addition, the medicinal qualities of this water were highlighted, it being recommended for *kidney, bladder, stomach, and intestine diseases*, as well as for *bronchial asthma, gout, or diabetes*. It is mentioned that the water is also good for daily use as drinking water. The ownership and marketing rights have been held by various companies or individuals, and the name of the water has always been centred on the chemical element *lithium*, and has been named Lythia, Lythinia, and Lithymus. In the 1960s, Berlescu et al. [34] and Binștoc [35] mentioned Tămășeu both for the hydromineral treatment of the excretory system and for the cure of digestive diseases with alkaline carbonated waters, which reduce uric acid in the blood. Also, Pricăjan [33] briefly discussed the chemical composition of Bihor's natural mineral waters, namely those from Tămășeu and Pădurea Neagră. Aniței et al. [36] consider Tămășeu a place with natural healing factors, and an exciting lowland bioclimate.

There are currently two boreholes at Tămășeu at a short distance from each other, located on the edge of the village near the ruins of the railway station, in the Barcău river meadow at about 107 m altitude. The borehole the water samples were taken from has a depth of 110 m, and the local administration replaced a segment of the column at a depth of 40–50 m and installed a new wellhead in 2003 [32]. This borehole is protected above ground by a closed concrete tube fitted with a metal pipe, through which water flows at low flow rates. The second borehole reaches up to 220 m depth, and water supply work

to the Parhida railway station is currently underway here. A methane gas pipeline and a storage station are located near the boreholes, and the surrounding farmland is cultivated with grain.

In the local development strategies of the Sălard commune, it is written that "before the world war the water from the borehole in Sîntimreu was bottled and it was useful for digestive diseases" [70] and that "living water has a therapeutic effect in the internal treatment of ailments of the digestive and hepatobiliary system" [71]. Aniței et al. only [36] mentioned Sîntimreu with mixed ferruginous, carbonated waters.

The current Sîntimreu borehole is located in the village near the school, at the base of the second terrace of the Barcău river at about 115 m in altitude. There were oak forests here in the past, but now these terraces are cultivated with vineyards. Due to modernization work, the borehole is now protected by a cover, and the water flows at high speed through two pipes with four outlets.

Currently, the natural mineral water from the Tămășeu—Sîntimreu—Sălard aquifer is unutilized and is discharged freely through the Tămășeu and Sîntimreu wells.

### 3.2. Physical-Chemical Indicators

3.2.1. Hydrochemical Characterization

The measured *pH values* for the analyzed waters fell within the limits specified by the legislative norms applied to both mineral waters and drinking water [5,6]. The analysis of the parameters reveals common aspects that give some particular characteristics to these waters. The pH values varied between a maximum value specific to the mineral water from Tămăseu (pH 7.5) to a minimum value found for the water from the Pădurea Neagră spring (slightly acidic, pH 6.5). The natural mineral water from Sântimreu has a pH value of 6.84 pH units. The lowest pH value determined in the water from the Pădurea Neagră spring is the result of a significant footprint of anions. Long-term consumption of water with a low pH can influence the solubility and bioavailability of some nutrients [2].

*Total mineralization* is a defining parameter that provides information about the potential use of the water, as well as the technical aspects that must be considered in exploitation. Related to this characteristic, Stoicescu [72] and Vernescu [73] mention that in Bihor County, there are a variety of acratic and hypertonic waters.

The quality of mineral waters depends on the concentration of dissolved mineral salts. Referring to this indicator, it was found that the highest mineralization (7128 mg/L) was obtained for the water of the borehole of Tămășeu in Barcăului Plain, and the lowest value was obtained for the water of the spring from Pădurea Neagră in the Plopișului Mountains (3086 mg/L) (Figure 3). According to Annex 3 to HG 1020/2005 [7] (with subsequent amendments and additions), and taking into account the content of mineral salts found, the natural mineral waters from the three localities fall into the category of highly mineralized natural mineral waters (residue content > 1500 mg/L). Highly mineralized waters are recommended to athletes and people performing physical activities at high temperatures, and are very useful to replenish the electrolytes lost through sweating.

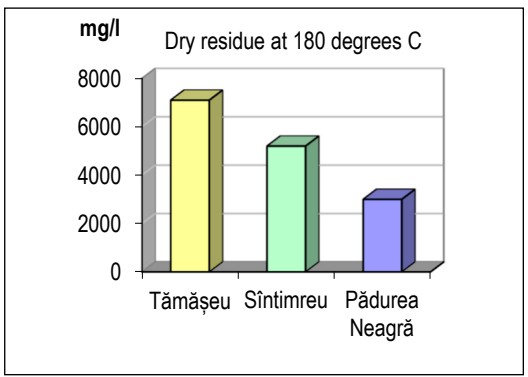

**Figure 3.** Dry residue at 180 °C.

According to Annex 3 of HG 1020/2005 [7] with subsequent amendments and additions, for a mineral water to be classified as acidulated, it is necessary to have a $CO_2$ *content* > 250 mg/L. Analyzing the results obtained for each mineral water, it was found that the mineral waters from Tămășeu, Sîntimreu, and Pădurea Neagră are acidulated, with the water from Pădurea Neagră spring having the highest $CO_2$ value (2004.2 mg/L) (Figure 4). In addition, since the acidulated mineral waters from Pădurea Neagră and Sîntimreu spontaneously and visibly released carbon dioxide, they can be classified as effervescent waters.

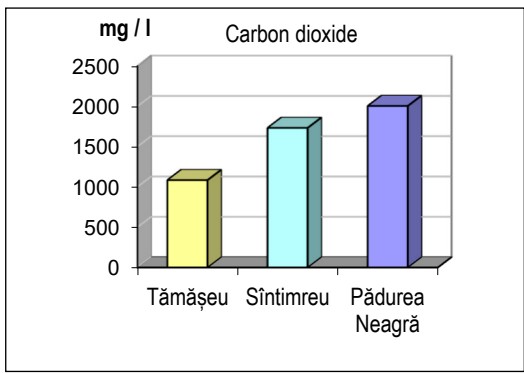

**Figure 4.** Carbon dioxide content.

3.2.2. Chemical Composition of Natural Mineral Waters

The chemical composition of mineral water, given by the specific content of dissolved salts, represents one of the main characteristics that can give it beneficial effects for health. The greatest non-uniformity in ion distribution is found in the cationic zone, where there are variations between the predominant characters of the mineral waters from the three locations. The sum of cations and anions dissolved in water, expressed in milliequivalents/liter, should be equal: ($\sum$cations) = ($\sum$anions). An imbalance of this equality can be attributed to errors in analytical determinations or electrolyte dissolution as a result of anthropogenic influences. The charts shown in Figure 5 give us clues about these aspects. The data presented in Figure 5 show that, among cations, calcium ($Ca^{2+}$), magnesium ($Mg^{2+}$), sodium ($Na^+$), and potassium ($K^+$) are predominant.

*Calcium* is predominantly found in the analyzed natural mineral waters, its origin being related to calcareous/dolomitic rocks. *Magnesium* is a cation present in natural mineral waters, usually together with calcium. It usually comes from dolomitic rocks. *Calcium* is an element with multiple functions in the human body, essential not only for the mineralization and health of the skeleton, but also for the proper functioning of the muscular system (regulates muscle contraction), the vascular system (mediates vasoconstriction and vasodilatation), and the nervous system (transmission of nerve impulses, ion exchange across cell membranes) [74,75]. Since adequate calcium intake is important for the body's health, there are many calcium-enriched food products and calcium supplements on the market. However, some supplements may have low absorption or be a source of gastric discomfort; therefore, mineral waters rich in calcium might be a potential source of calcium. The bioavailability of calcium from calcium-rich mineral waters is comparable or slightly higher than that of calcium from milk, with mineral waters providing over 40% of the recommended daily intake of calcium [75–77]. The consumption of mineral waters, especially alkaline mineral waters rich in bicarbonates and calcium, has a positive impact on bone mass and bone density, as well as in the prevention of osteoporosis. Therefore, calcium-rich mineral waters can be recommended as calcium supplements [74,75,77]. *Magnesium* is involved in various physiological processes, such as mitochondrial integrity, adenosine-triphosphate function, DNA synthesis, bone mineralization, or muscle activity [78]. Magnesium has a cardioprotective effect, and drinking hard mineral waters rich in magnesium and calcium reduces the risk of mortality from cardiovascular diseases [79].

These two cations have higher concentrations in the water from Sîntimreu and Pădurea Neagră (Figure 5).

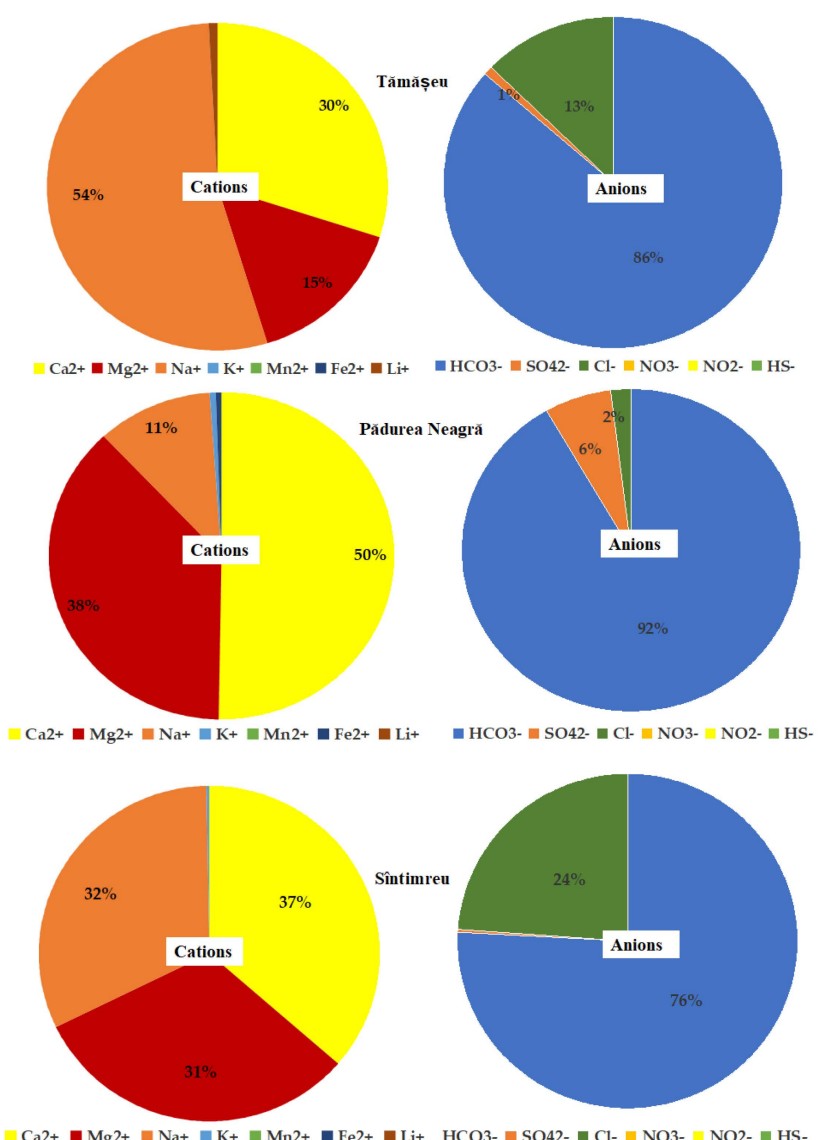

**Figure 5.** Ionic balances of waters from Tămășeu, Sîntimreu, and Pădurea Neagră.

*Sodium*, an essential element for human life, characterized by high reactivity, and can be found in most groundwater. Concentrations higher than 200 mg/L can influence the taste of water, which is not the case for the waters analyzed in this study, since the concentration of this cation was in the range of 47.8 mg/L (Tămășeu), 45.3 mg/L (Sîntimreu), and 9.2 mg/L (Pădurea Neagră). In general, sodium salts have no acute toxicity due to the ability of the healthy mature kidney to excrete sodium [6]. *Potassium* poisoning by ingestion is rare due to the kidney's ability to excrete it rapidly in the absence of pre-existing kidney damage, and large single doses usually induce vomiting [6]. Potassium was found in higher amounts in the water from Pădurea Neagră spring (1.35 mg/L).

*Lithium* in the mineral waters is related to the presence of clay muds where it was retained during its underground circulation. Lithiniferous waters are considered to be mineral waters with a lithium concentration of more than 3 mg/L [37]. In Romania, there are several mineral waters rich in lithium, such as the Cașin-Iacobeni spring (Perla Cașinului water, lithium concentration of 6.6 mg/L, the richest lithium water in Romania), the Harghita spring (6.2 mg/L lithium), the Maria spring from Malnaș-Băi (5.6 mg/L), and

the Matilda spring from Bodoc resort (5.3 mg/L) [5,12,80]. Among the waters under study by us, the highest concentration of lithium (4.4 mg/L) was determined in the mineral water from the Tămășeu borehole, which can fit this water into the category of lithiniferous waters. Lithium salts (e.g., lithium carbonate) are implicated in the treatment of bipolar disorder, but these drugs have a low therapeutic index and multiple side effects [81]. The bioavailability of lithium in mineral waters is very high, and studies showed that drinking lithium-rich mineral waters can reduce the risk of suicide and prevent neurological disorders such as dementia or Alzheimer's disease [12,81].

*Iron* has an important role in the body. Ferruginous waters are administered as internal crenotherapy, freshly brought from the spring and during meals, because iron is inactivated in contact with air. Only bivalent iron is active and is absorbed in the presence of hydrochloric acid and vitamin C. Ferruginous waters are implicated in iron deficiency anemia, gastric achilia, gastric surgery, etc. [12]. Iron was found in significant amounts in the sample taken from Pădurea Neagră spring (0.57 mg/L), which is highlighted in Figure 5.

Regarding the anions, the preponderance of the bicarbonate anion ($HCO_3^-$) was found, followed by chloride ($Cl^-$) and sulphate ($SO_4^{2-}$) anions. The predominance of bicarbonate anions over those of strong acids (chlorides or sulphates) was found in all analyzed mineral waters. The bicarbonate ion is produced when water reacts with carbonate rocks such as limestone and dolomite. The mineral water from the Pădurea Neagră spring has the highest concentration of bicarbonate, followed by that from Tămășeu and Sîntimreu. An appreciable amount of sulphates were found in the water from the Pădurea Neagră spring (Figure 5).

Mineral waters rich in sodium or potassium bicarbonates can neutralize gastric acid, thus increasing the pH value of the stomach and reducing heartburn in adult patients. Considering the high tolerability, such mineral waters might be considered as alternative to drug treatments. In addition, depending on the time of administration (before or during the meal), carbonated alkaline waters can reduce or stimulate gastric secretions, influencing gastrointestinal motility and gallbladder activity [11,12,82,83]. Alkaline-earth waters contain the bicarbonate anion bound to calcium or magnesium cations, and their main applications are for digestive ailments (gastritis, colitis, gastric or duodenal ulcers, chronic enterocolitis, etc.). Since many of them have calcium as the main mineral, they are also effective in treating rickets and allergies [12]. Mineral waters rich in salts (e.g., bicarbonate, chlorine, sulphate, magnesium, and sodium) not only increased intestinal transit through laxative or purgative activity with beneficial effects for constipation, but also stimulated gallbladder contraction and bile acid elimination through stool, and decreased the level of serum total cholesterol and LDL-cholesterol. This therefore might be considered an alternative to hypocholesterolemic drugs [84–86]. In addition, bicarbonate-sulphate-alkaline-earth mineral waters have antitoxic and trophic effects on the liver [16].

A particular case is the water from Sîntimreu, with a strong smell of hydrogen sulphide. In addition to the effects determined by the existing cations and anions, hydrogen sulphide contributes to the properties of this water. Sulphurous waters stimulate gastric and bile secretion through choleretic and cholecystokinetic effects. Hydrogen sulphide protects the intestinal flora from various chemical attacks and therefore heals mucosal wounds and reduces local inflammation. Its cytoprotective activity, local vasodilation, and ability to promote new blood vessel growth also contributes to these effects. Sulphur is a constituent of some amino acids (e.g., cysteine), and through the thiol group of hydrogen sulphide, crenotherapy with sulphurous waters can heal lesions of the respiratory tract mucosa (bronchitis, chronic rhinitis, COPD) [11,19,87]. In addition, sulphurous waters have hypoglycemic and antioxidant properties. Both the antidiabetic and antioxidant effects are mainly due to hydrogen sulphide, and are beneficial in metabolic diseases such as diabetes [22,23].

In order to classify the natural mineral waters from the three localities from a hydrochemical point of view, the predominant ions ($Na^+$, $K^+$, $Ca^{2+}$, $Mg^{2+}$, $HCO_3^-$, $Cl^-$, $SO_4^{2-}$)

were represented on the Piper diagram (Figure 6). The diagram shows the concentrations of the majority of cations and anions arranged on two separate trilinear diagrams, together with a central diagram (diamond) in which the points from the two trilinear diagrams are projected. In the case of the mineral water from the Pădurea Neagră spring, the location is found in the quadrant represented by the majority cation $Ca^{2+}$ (calcium type). The natural mineral water sampled from Sîntimreu belongs to the specific area "no dominant type". The natural mineral water from Tămășeu is located in the quadrant with predominantly alkaline ions (sodium and potassium type). From an anionic point of view, all three types of natural mineral waters analyzed are in the area specific to the majority bicarbonate ion (bicarbonate type) (Figure 6).

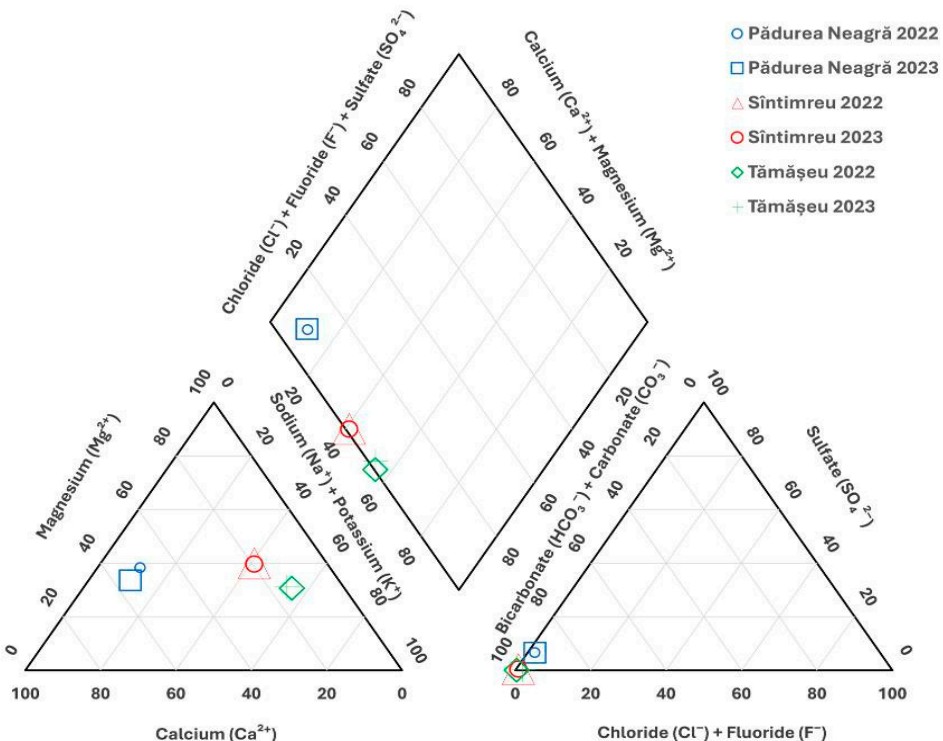

**Figure 6.** The Piper diagram of the studied natural mineral waters.

Mineral waters were classified according to the dominant constituents. In Figures 6 and 7 is the sum of cations $Ca^{2+} + Mg^{2+} > Na^+ + K^+$, and the sum of anions $HCO_3^- + CO_3^{2-} > Cl^- + SO_4^{2-}$. The dominant cations are the alkaline-earth ones, and the dominant anion is bicarbonate.

As a general characteristic of the natural mineral waters analyzed in this study, according to Piper and Stiff representations (Figures 6 and 7), the hydrochemical fingerprint is as follows:

● The mineral water from Tămășeu is predominantly sodium bicarbonate (Na-$HCO_3^-$),
● The mineral water from Sîntimreu is mixed calcium-magnesian bicarbonate type (Ca-Mg-$HCO_3^-$),
● The mineral water from Pădurea Neagră is predominantly calcium bicarbonate (Ca-$HCO_3^-$).

The dominant hydrochemical type is specific to the aquifers in the three analyzed areas due to the fact that most of them are concentrated in reservoirs rich in carbonate. The mineralization of an underground water source can show variations within the specific limits of the lithology of the location area, and according to the literature, these variations can fall within the range of 15–25%.

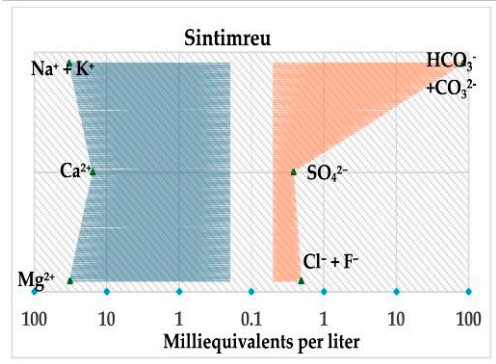
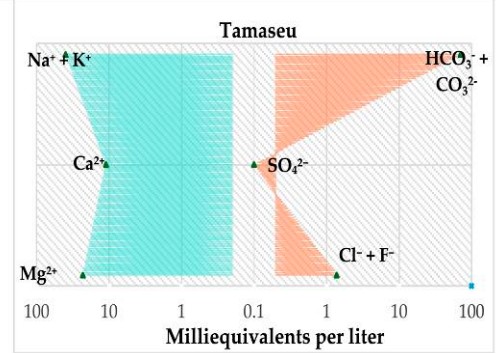
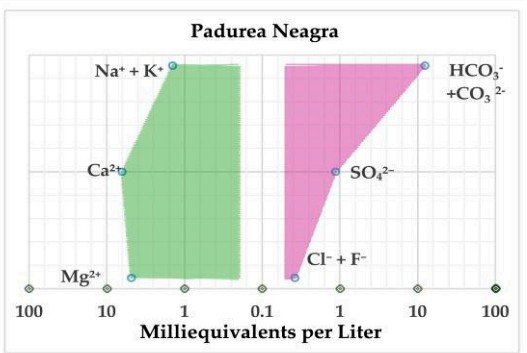

**Figure 7.** The Stiff diagram of the studied natural mineral waters.

### 3.3. Microbiological Characterization

In order to microbiologically characterize the natural mineral water samples from Tămășeu, Sîntimreu, and Pădurea Neagră, six indicators during the sampling campaign (2022–2023) were determined. As can be seen in Table 3, there are no pathogenic microorganisms that might represent a danger to human health in the studied samples. According to the national legislation for mineral waters, the reference values are 20 germs/mL at a temperature of 20–22 °C in 72 h and 5 germs/mL at a temperature of 37 °C in 24 h; therefore, the waters analyzed in this study comply with the quality conditions imposed by both drinking water and mineral-water-specific legislation.

**Table 3.** Microbiological indicators specific to mineral waters from Tămășeu, Sîntimreu, and Pădurea Neagră.

| Microbiological Indicators | Unit | Results | | | HG 1020/2005 (Government Decision) | Law 458/2002 |
| --- | --- | --- | --- | --- | --- | --- |
| | | Tămășeu | Sîntimreu | Pădurea Neagră | | |
| *The total colony count (CFU) at 22 °C* | CFU/mL | No abnormal changes | No abnormal changes | No abnormal changes | 100 | 100 |
| *The total colony count (CFU) at 37°C* | CFU/mL | No abnormal changes | No abnormal changes | No abnormal changes | 20 | 20 |
| *Total Coliform bacteria and Escherichia coli* | CFU/250 mL | 0 | 0 | 0 | absent | 0 |
| *Intestinal enterococci* | CFU/250 mL | 0 | 0 | 0 | absent | 0 |
| *Pseudomonas aeruginosa* | CFU/250 mL | 0 | 0 | 0 | absent | 0 |
| *Clostridium perfringens* | CFU/50 mL | 0 | 0 | 0 | absent | 0 |

### 3.4. Analysis of the Questionnaire

Below, we present the analysis of the questionnaire regarding *the residents' perception on the quality and effect on health of the natural mineral waters* from the local boreholes/springs under study (Tămășeu, Sîntimreu and Pădurea Neagră).

### 3.4.1. The Quality and the Effect of Mineral Waters on Health

Regarding the main source of drinking water of the inhabitants of the three localities and neighboring area who answered the questionnaire, statistical analysis shows that most of them (28%) use bottled water from the store, 20% of the respondents use the mineral water from the borehole in Sîntimreu, 18% use the mineral water spring from the holiday village Pădurea Neagră, and 18% use the water from the well dug in the yard of their house. Only 6% of the respondents use the mineral water from the borehole in Tămășeu village (Figure 8).

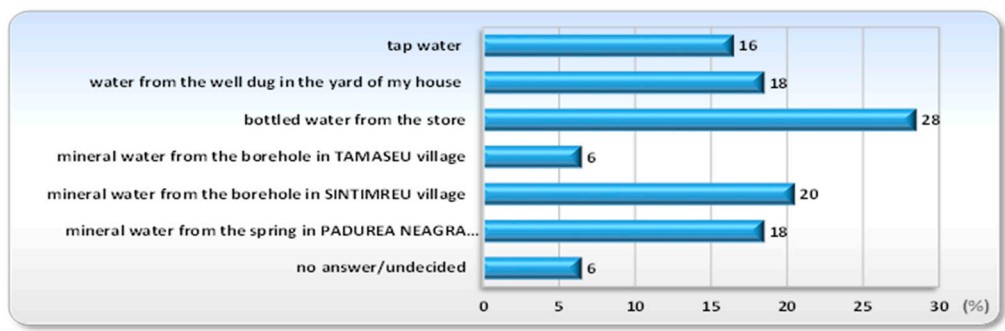

**Figure 8.** Distribution of respondents according to their main drinking water source.

The reason could lie in the fact that the borehole in Sîntimreu is easily accessible to the local population, being located near the center of the village and having a large water flow. On the other hand, the one in Tămășeu is located outside the village, about 500 m away, and the water flow is low. The use of water from the Tămășeu borehole in such a small percentage is also attributed, by some of those surveyed, to the gastrointestinal problems it causes (e.g., diarrhea). Many of the people questioned from the village of Tămășeu and the neighboring village of Parhida stated that they had not drunk mineral water from the borehole for several years. The current source of drinking water for those surveyed in Tămășeu village is either bottled water from the store or water from the local drinking water network (tap water). In Parhida, apart from the bottled water from the store, the respondents also use water from the borehole in the center of the village extracted from about 25 m deep (called "artesian" by the locals).

When asked what makes them choose a bottled water brand over local mineral water from spring/borehole when shopping, 38% of the respondents answered the taste, 24% the origin of the water source, 22% did not choose any answer, 16% the chemical composition of the water, and only 4% answered the advertisement. Some of them also specified the price as important in choosing the brand of bottled water.

Issues affecting the quality of mineral water from the local borehole/spring are viewed differently by the respondents. Most of them (46%) consider that the local mineral waters have no quality problems, 14% of the respondents believe that iron affects the water quality since it is at too high of a concentration, and 12% of them noted the taste or smell, respectively limestone (Figure 9). The fewest of the respondents consider salinity to be a problem (2%), solid particles in the water, or nitrates/nitrites (4%).

Regarding the frequency of mineral water consumption, most respondents (32%) drink mineral water daily from the local spring/borehole, 22% of them occasionally drink mineral water, 16% drank it once a month, 14% rarely drank it, and the fewest respondents drank it once every 2–3 days (2%) or once a week (6%) (Figure 10).

When asked how much mineral water they drink per day, most respondents (34%) did not give any answer, as many of them occasionally or rarely drink mineral water from the local borehole/spring. In total, 24% of the respondents consume more than 1 L/day, 16% about 1 L, and the rest of the respondents consume less than 1 L.

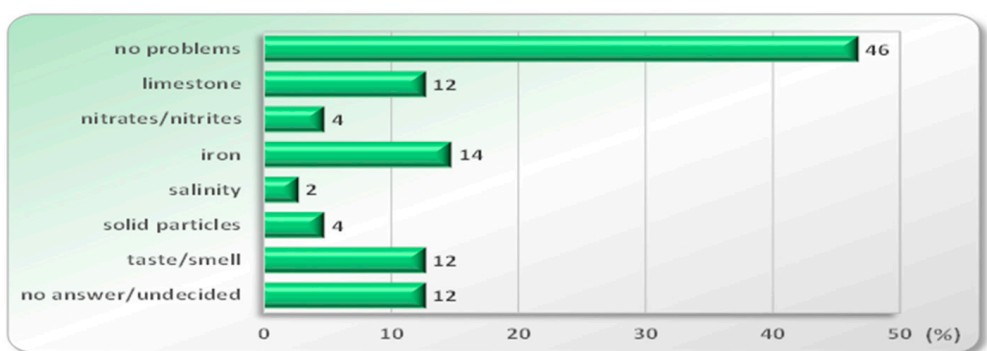

**Figure 9.** Respondents' perception on issues affecting the quality of mineral water.

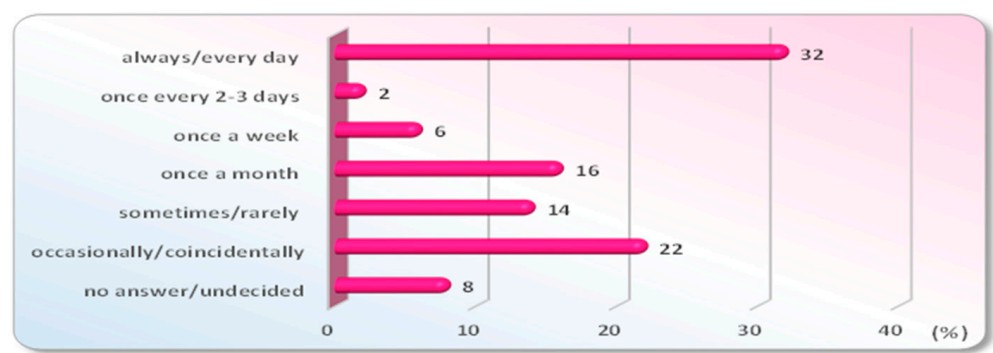

**Figure 10.** Distribution of respondents according to the frequency of mineral water consumption.

When asked how long they have been consuming mineral water from the local borehole/spring, most respondents (24%) said since childhood, 22% of respondents gave no answer, and 20% of them stated that they have been drinking mineral water for 10 years or less. Some respondents stated that they have been drinking the local mineral water for over 40, 50, or even 60 years.

When asked how long the mineral water stays clear, 38% of the respondents answered a few days, and 36% answered one day. Only 14% answered a few hours. After this period of time, deposits appear on the bottom of the container. Questioning the population, we found out that some locals from the village of Pădurea Neagră do not consume the mineral water from the spring in the holiday village precisely because of the deposits that occur on the bottom of the containers after a few days. As a result, they prefer to drink water from the village spring (spring water), which keeps its clarity and does not show deposits. The precipitation of some mineral salts is also present at the two boreholes (Tămășeu and Sîntimreu).

Asked if they add something to the mineral water in order not to change its color, taste or smell, the majority of respondents (70%) stated that they do not add anything, as they do not mind the organoleptic characteristics of the water. Additionally, 22% did not give any answer, and only 8% said they add lemon salt (or apple cider vinegar).

The majority of those surveyed (76%) believe that the mineral water from the local borehole/spring is much more hygienic/cleaner than the drinking water from the tap or the water from the well in the yard. Only 12% believe that it is not cleaner.

The majority of respondents (86%) stated that they do not suffer from any medical condition that requires them to drink more water than usual. Only 6% stated that they need to drink more water. For example, some of those surveyed consume more mineral water from the Sîmtimreu borehole when they have stomach aches.

The majority of those surveyed (70%) answered negatively when asked if they suffer/suffered from any acute condition for which the internal (drinking) or external (bathing) use of mineral water from the local borehole/spring made them feel an improvement in

their health. Only 12% answered in the affirmative, mentioning that they suffer/suffered from conditions such as gastritis, gastric ulcers, flatulence, or liver conditions.

80% of the respondents answered negatively when asked if they suffer from any chronic condition for which the internal (drinking) or external (bathing) use of mineral water from the local borehole/spring makes them feel an improvement in their health. Only 8% answered affirmatively, declaring that the disease they suffer from is gastritis or gastric ulcers.

When asked how they feel after drinking mineral water from the local borehole/spring, 38% of respondents did not give any answer, 28% of them declared that mineral water gives them a feeling of well-being, good mood, and vitality (they do not feel melancholy, depressed, or angry), 20% of them answered that they felt more energetic, and 20% of them said that their digestive problems decreased (the activity of the digestive system improved). Smaller percentages of those surveyed show that their urinary tract problems (2%) and biliary disorders (2%) have decreased, or their cardiovascular system activity (2%) has improved (Figure 11). Some respondents stated that their liver conditions decreased. As shown above, some people questioned stated that they do not consume mineral water (especially the water from the Tămășeu borehole) because it causes them gastric disorders.

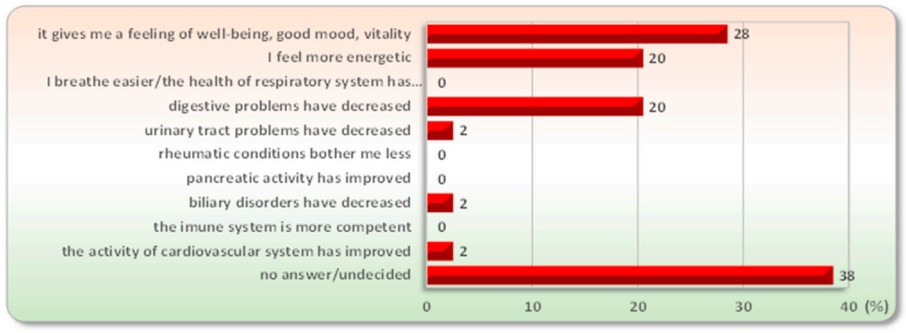

**Figure 11.** The effect on health perceived by the population after drinking mineral water.

Most of the surveyed people do not use the mineral water from the local borehole/spring in the kitchen for various reasons: 32% because the borehole/spring is too far and they only use the mineral water for drinking, 20% because the mineral water changes the taste of food/drinks, and 18% do not use it for other reasons (Figure 12). A total of 14% did not give any answer. Only 12% of respondents use the mineral water for food preparation, and a smaller percentage of them for the preparation of alcoholic (spritz, etc.) or non-alcoholic (juices, herbal infusions, etc.) drinks. The locals use the mineral water from the local borehole to make spritz, especially in Sîntimreu, where there are famous vineyards.

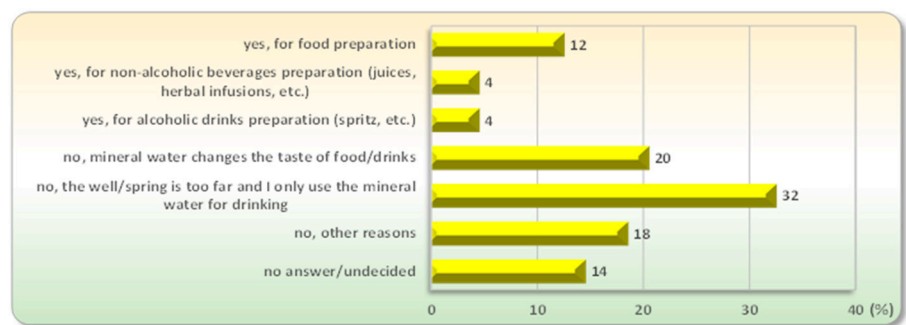

**Figure 12.** Distribution of respondents according to the use of mineral water in the kitchen.

Most of those surveyed (64%) drank mineral water from the local borehole/spring during the Covid pandemic (2020–2022), and 32% did not. Of the 64% respondents who

affirmatively answered to the previous question, half (52.2%) stated that the mineral water did not help them not get infected with the SARS-CoV-2 virus or to recover faster, and only some of them (8.7%) stated that the water helped them. A total of 39.1% did not answer.

3.4.2. The Socio-Demographic Situation of the Respondents

Regarding *the socio-demographic situation* of the respondents, a fairly uniform distribution—in terms of their residence—between the three locations analyzed (Tămășeu, Sîntimreu, Pădurea Neagră) can be observed. Most respondents live in Pădurea Neagră and its surrounding localities (39%), followed by those in Sîntimreu and the neighboring villages (32%), then by those in Tămășeu and the neighboring localities (29%). The questionnaire was filled out by respondents from a total of 12 localities: two cities (Oradea and Marghita) and ten villages (Tămășeu, Parhida, Sălard, Sîntimreu, Mihai Bravu, Pădurea Neagră, Voivozi, Popești, Ciutelec, Mișca).

A fairly uniform gender distribution of those who answered the questionnaire can be observed (56% women and 44% men). Analyzing the age of the respondents (Figure 13), it can be seen that most fall into the age category 51–60 years (28%), followed by the 21–40 years category (24%), then the 41–50 years category (20%). Therefore, half of the respondents are aged between 41–60 years, and 20% of them are over 60 years old.

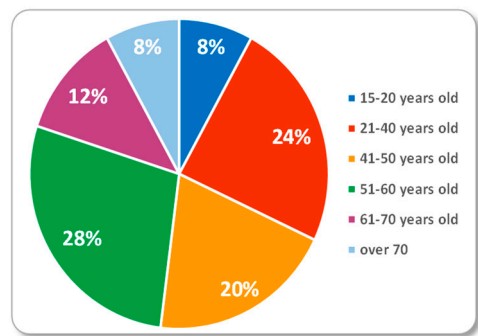

**Figure 13.** Distribution of respondents according to age.

Regarding the level of education of the respondents (Figure 14), 61.2% of them have a university education (22.4% master's studies, 20.4% bachelor studies, 18.4% doctoral studies), 28.6% have high school education, and 10.2% only have secondary school education. This proportion is explained by the fact that many respondents are educated young people or adults who filled out the questionnaire online. The elderly rural population (over 60 years old), with less education, does not use the Internet.

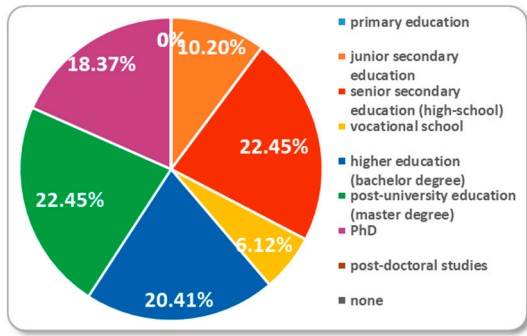

**Figure 14.** Distribution of respondents according to the level of education.

Most of those surveyed are permanent employees (69.4%), followed by retirees (14.3%). Temporary employees, pupils and students together represent 12.2% of those surveyed. Other social categories show very small percentages (administrators, farmers).

The face-to-face investigation helped us to report special cases, such as that of the mayor of Sălard commune, Mr. Nagy Miklós, who benefited greatly from the healing effect of the mineral water from the Sîntimreu borehole. He told us that in his youth, he underwent emergency surgery due to a perforated gastric ulcer, his medical condition being critical and having a minimal chance of survival. At the doctor's recommendation, he drank at least 2 L of mineral water/day from that borehole for about 3 months, and his health improved significantly. He stated that he was cured of gastric ulcers, and since then, he drinks that water daily.

When it is used as an internal cure (crenotherapy), the mineral water from the boreholes in Sîntimreu and Tămășeu initially causes gastrointestinal disorders (e.g., diarrhea) for about a week. After that, this effect disappears. It is possible that this effect is a result of the detoxification of the body, which the Lithinia Company also talks about on its website [88].

Several locals from Sîntimreu told us that there are many people over 90–95 years old (even over 100 years old—declared by the mayor of the Sălard commune) in their village. People from this area are long-lived, and one of the reasons might be drinking the local mineral water (another reason for their longevity could be drinking the local wine). The inhabitants of Sîntimreu village do not want the mineral water to be bottled. They do not want the local mineral water to be bottled and become the property of some company that would limit their access to water, meaning they would have to buy it from the store.

This study is not exhaustive and has some limitations, such as the sampling of a small number of samples, and the inability, at this time, to extend the study to the determination of other characteristics specific to natural mineral waters. Regarding the public opinion on the potential impact of these waters on health, we encountered some difficulties in communicating with the inhabitants of various ethnicities in the studied area. We also note the difficultness of speaking directly to all study participants. However, this study represents an important step for identifying new opportunities for the exploitation and management of natural mineral water sources in Bihor County, in the context of a sustainable management of this resource. In addition, through the questionnaire and discussions with the residents and local authorities, we think that we have taken a relevant step in the process of raising awareness in the population regarding the benefits that these natural mineral waters can bring for the therapy of various ailments.

The mineral waters from Tămășeu, Sîntimreu and Pădurea Neagră have microbiological and physical–chemical characteristics that comply with the current legislation specific to both drinking water and natural mineral waters, and our future research will aim to evaluate all the characteristics specified in HG 1020/2005 for including the water sources from Tămășeu, Sîntimreu and Pădurea Neagră in the category of natural mineral water sources recognized in Romania.

## 4. Conclusions

The microbiological and physical-chemical characteristics studied give us the opportunity to state that the waters from Tămășeu, Sîntimreu, and Pădurea Neagră represent a potential source of natural mineral water. The analysis of the questionnaire helped us to reveal people's opinion about these waters. Many respondents consider that the studied waters do not have quality problems and have beneficial effects, especially regarding gastrointestinal ailments, which could pave the way for an in-depth medical study on the therapeutic potential of these waters; this is one of the objectives of our future studies. Obtaining additional data for a detailed chemical characterization of these natural mineral water sources is another future goal. Taking into account the medical studies that show the high bioavailability of minerals from various mineral waters, we can consider these waters as accessible sources of calcium, magnesium, sodium, iron, bicarbonate, and sulfates with beneficial effects in the prevention or treatment of various diseases. A particular case is the mineral water from the Tămășeu borehole with a high lithium content, which might be recommended as complementary therapy in the treatment of psychiatric disorders. In

addition, the water from Sîntimreu, due to its hydrogen sulphide content, could have beneficial effects for various pathologies. Correlated with historical data, our results indicate the hydraulic, nutritional, and therapeutic potential of these natural mineral waters, which is a benefit for the local population. The present study could increase the interest of the scientific community for the in-depth study of these waters and their inclusion on the list of natural mineral water resources in Romania. Located in an area with multiple types of hydromineral and hydrothermomineral resources with varied physical–chemical characteristics, Bihor County can offer new opportunities for their hydrological and therapeutic exploitation.

**Author Contributions:** Conceptualization, R.L., E.P., E.Ș., G.S. and A.-P.C.; Methodology, R.L., E.P., E.Ș., G.S. and A.-P.C.; Validation, R.L., E.P., E.Ș., G.S. and A.-P.C.; Investigation, E.P., E.Ș., G.S., A.-P.C. and R.L.; Writing—original draft preparation, R.L., E.P., E.Ș., G.S. and A.-P.C.; Writing—review and editing, R.L., E.P., E.Ș., G.S. and A.-P.C. All authors have read and agreed to the published version of the manuscript.

**Funding:** Publishing of this research was funded by University of Oradea.

**Institutional Review Board Statement:** Not applicable.

**Informed Consent Statement:** Not applicable.

**Data Availability Statement:** The data sets supporting the conclusions made are included in this article.

**Conflicts of Interest:** The authors declare no conflict of interest.

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
