# Peer review of "Hydrochemical and Microbiological Investigations and the Therapeutic Potential of Some Mineral Waters from Bihor County, Romania"

_sustainability, doi:10.3390/su152115640_

Round 1
Reviewer 1 Report
Comments and Suggestions for Authors
General Comments
This study entitled “Hydrochemical and Microbiological Investigations and the Therapeutic Potential of Some Mineral Waters from Bihor County, Romania”, authored by Linc et al. is a very interesting work. It analyses the hydrochemical and microbiological properties of mineral water and considers the sociological perspective as well. The authors analyse their findings very well; however, some bibliographical justification is needed.
Specific comments
Line 1: please specify your submission type
Line 19: what kind of analysis and of what? please articulate and rephrase the lines 19-20
Line 20: please add the country name : Bihor country(Romania)
Line 21-22: why you use the terms sample/year ? if not necessary please remove them
Line 36: “Water is the source of our existence on Earth” this statement is not accurate. Please rephrase with something like this: “water is on of the main sources of our …”
Line 40: please don’t generalize your statements. Remove the words “every” and rephrase.
Line 43: please mention some of these “beneficial properties of these waters for human health” and add the associated references.
Line 42: please check the citation style and add the Authors names when necessary. Also, the same for line 49 and throughout the text.
Line 57: what is this distinction between "natural mineral waters" and "medicinal waters”. Please analyse it briefly.
Line 61: the words empirical data are not appropriate. Please remove it
Line 69: please define the term crenotherapy
Line 81: “make a substantial contribution” please rephrase with “ might make a substantial contribution”
Line 82: “of side effects of drugs” please rephrase with “of possible side effects of drugs”
Line 89: “be easily absorbed through the skin and mucosa and gives systemic effects” reference is needed for this statement
Lines 91:93: “ During hydropinotherapy with sulphurous mineral waters, there is an increase in the secretion of insulin leading to the burning of glucose.” reference is needed for this statement
Line 201: CFU need to explain this abbreviation
Line 315: please specify these limits here
Line 342: better analysis figures 3& fig.4 are needed
Line 354-357: “also for the proper functioning of the muscular system (regulates muscle contraction), the vascular system (mediates vasoconstriction and vasodilatation) and the nervous system (transmission of nerve impulses, ionexchange across cell membranes).”reference are needed here to justify each of these systems function
Line 361-363: “The bioavailability of calcium from calcium-rich mineral waters is comparable or slightly higher than that of calcium from milk, with mineral waters providing over 40% of the recommended daily intake of calcium” please justify with reference for this statement
Line 366-368: Magnesium is involved in various physiological processes, such as mitochondrial integrity, adenosine-triphosphate function, DNA synthesis, bone mineralization or muscle activity: please add the relevant reference.
Line 377: figure 5 needs better analysis. Also the size of pies charts can be reduced.
Line 384-385: healthy mature kidney to excrete sodium. Potassium poisoning by ingestion is rare due to the kidney's ability to excrete it rapidly in the absence of pre-existing kidney damage: please add the necessary references to justify these
Line 397: but these drugs have a low therapeutic index and multiple side effect: Please justify here with a reference
Line 421-422: have calcium as the main mineral, they are also effective in rickets and allergies: reference is needed.
Line 444: figure 6 needs better analysis. The same for figures 7-12.
Line 506-507: please correct the brackets. The same for the rest text
Reviewer 2 Report
Comments and Suggestions for Authors
The paper is of good scientific quality, it provides preliminary work on the therapeutic potential of mineral waters from Bihor Country in Romania. Some aspects concerning human health should be investigated further in other papers, in the conclusion, the authors should highlight this aspect better. To clarify the instability of the water and its deposit is another point to investigate related to human safety.
Some minor modifications are necessary:
Fig.1 It is not clear the relationship between the principal figure and the inset
Fig.7 The font characters in the figure (name and units) are different in the three figures: they must be the same for clarity.
Reviewer 3 Report
Comments and Suggestions for Authors
Acceptable
Author Response
Thank you for your acceptance.
Reviewer 4 Report
Comments and Suggestions for Authors
Authors presented "Hydrochemical and Microbiological Investigations and the Therapeutic Potential of Some Mineral Waters from Bihor
County, Romania" in this paper. The different investigations of hydrochemical and microbiological biomolecules and minerals of water are presented. The manuscript is interesting but needs to be revised extensively to be applicable to this journal.
1. The research related to this topic needs to be incorporated in the comparative table to show how the advancement has been done in this topic. A comparison with the method is provided which is not enough.
2. The paper needs to incorporate more research papers to make the manuscript stronger.
3. More analysis needs to be put in the manuscript to incorporate details about microbiological and hydrological investigations.
4. The English of the manuscript can be improved for better readership.
5. Fig. 5, 6, 7. need to be improved as the font size is not visible. Please look into all the figures and increase the font size for better readership
6. A Pi chart should be presented for different applications.
Overall, the manuscript needs to be revised as per the given comments to be accepted in this journal.
Comments on the Quality of English LanguageEnglish grammar need to be improved for better readership
Round 2
Reviewer 4 Report
Comments and Suggestions for Authors
The paper can be accepted in its current form
Comments on the Quality of English LanguageThe paper can be accepted in its current form